# Platelet-Membrane-Encapsulated Carvedilol with Improved Targeting Ability for Relieving Myocardial Ischemia–Reperfusion Injury

**DOI:** 10.3390/membranes12060605

**Published:** 2022-06-10

**Authors:** Tingting Zhou, Xuechao Yang, Tianyi Wang, Mingming Xu, Zhanghao Huang, Runze Yu, Yi Jiang, Youlang Zhou, Jiahai Shi

**Affiliations:** 1Nantong Key Laboratory of Translational Medicine in Cardiothoracic Diseases, Research Institution of Translational Medicine in Cardiothoracic Diseases, Affiliated Hospital of Nantong University, Nantong 226001, China; 2013320117@stmail.ntu.edu.cn (T.Z.); yxc6688@ntu.edu.cn (X.Y.); 2013510026@stmail.ntu.edu.cn (T.W.); 1931320140@stmail.ntu.edu.cn (M.X.); 1931320141@stmail.ntu.edu.cn (Z.H.); 2Department of Thoracic Surgery, Affiliated Hospital of Nantong University, Nantong 226001, China; 3Nantong University Medical School, Nantong 226001, China; 2013310096@stmail.ntu.edu.cn; 4Department of Pain, Affiliated Hospital of Nantong University, Nantong 226001, China; 1931320258@stmail.ntu.edu.cn; 5Research Center of Clinical Medicine, Affiliated Hospital of Nantong University, Nantong 226001, China; 6School of Public Health, Nantong University, Nantong 226019, China

**Keywords:** myocardial ischemia–reperfusion injury (MIRI), platelet membrane vesicles (PMVs), Carvedilol, targeted delivery treatment

## Abstract

In recent years, cell membrane drug delivery systems have received increasing attention. However, drug-loaded membrane delivery systems targeting therapy in myocardial ischemia–reperfusion injury (MIRI) have been relatively rarely studied. The purpose of this study was to explore the protective effect of platelet-membrane-encapsulated Carvedilol on MIRI. We extracted platelets from the blood of adult SD rats and prepared platelet membrane vesicles (PMVs). Carvedilol, a nonselective β-blocker, was encapsulated into the PMVs. In order to determine the best encapsulation rate and drug-loading rate, three different concentrations of Carvedilol in low, medium, and high amounts were fused to the PMVs in different volume ratios (drugs/PMVs at 2:1, 1:1, 1:2, and 4:1) for determining the optimum concentration and volume ratio. By comparing other delivery methods, including abdominal injection and intravenous administration, the efficacy of PMVs-encapsulated drug-targeted delivery treatment was observed. The PMVs have the ability to target ischemic-damaged myocardial tissue, and the concentration and volume ratio at the optimum encapsulation rate and the drug-loading rate are 0.5 mg and 1:1. We verified that PMVs@Carvedilol had better therapeutic effects compared to other treatment groups, and immunofluorescence observation showed a significant improvement in the apoptosis indicators and infarction area of myocardial cells. Targeted administration of PMVs@Carvedilol may be a promising treatment for myocardial reperfusion injury, as it significantly improves postinjury cardiac function and increases drug utilization compared to other delivery methods.

## 1. Introduction

Myocardial infarction is a type of irreparable damage caused by severe and persistent myocardial ischemia, usually caused by thrombosis and vascular blockage due to coronary atherosclerosis [1,2]. The only way to salvage the ischemic myocardium of a myocardial infarction is to perform timely reperfusion. However, reperfusion not only saves the ischemic myocardium but also induces another type of irreversible injury, which is known as MIRI, a condition in which the restoration of the blood supply to the myocardium after ischemia leads to metabolic dysfunction and structural damage [1,3,4]. Currently, it is generally accepted that the mechanism of MIRI is related to apoptosis, oxidative stress, intracellular calcium overload, endothelial and inflammatory responses, mitochondrial damage, and impaired myocardial energy metabolism [5,6,7,8].

Platelet adheres to coronary endothelial cells cause acute thrombosis in the formation of ischemia–reperfusion injury, so antiplatelet therapy is now an important treatment modality [9]. Platelets are small pieces of cytoplasm that are shed from the cytoplasm of mature megakaryocytes in the bone marrow and play a critical role in various physical activities within the body, including coagulation, hemostasis, natural immune responses, and cancer metastasis [7,10,11]. Platelet and platelet analog vectors have many advantages compared to conventional polymers, liposomes, and inorganic nanoparticles. PMVs are a type of cell membrane that itself has a potential camouflage effect that allows it to escape immunogenic monitoring, with low systemic cytotoxicity, to extend blood circulation [12,13,14]. Platelet-mediated delivery of tumor-targeted drugs is currently a popular topic of research, including the use of platelets to target lymphoma with platelet-encapsulated DOX, which enhanced the inhibition of Raji cell growth and reduced the toxicity of DOX to the heart [15,16]. However, in addition to research in tumor targeting, furthermore, other types of diseases, such as hematological disorders and cardiovascular and cerebrovascular diseases, can be targeted through the platelet membrane delivery of drugs or binding molecules [17,18,19]. In addition, the expression of useful proteins elicited by the targeting of platelets can be exploited to treat a number of bleeding-related genetic disorders [20]. For atherosclerotic diseases, designing platelets to encase specific nanoparticles and targeting them to atheromatous plaques allows for a better assessment of disease progression [21]. Of course, in the field of myocardial ischemia–reperfusion injury, there are also studies using platelets or platelet membrane nanocarriers encapsulated with rt-PA to directly dissolve thrombi and reduce fibril formation for therapeutic purposes [22]. It can therefore be argued that platelet-targeted therapy can target not only the site of ischemia but any immune response, and cancer metastasis can be further investigated via its natural properties [23,24,25,26,27].

Carvedilol (CVD) is a third-generation nonselective β-blocker that also has an α-blocking effect [28,29,30]. It is widely used in the clinical treatment of hypertension and heart failure, and it can significantly improve the prognosis of heart failure patients [31,32,33,34,35,36]. The mechanism of action includes antioxidant effects, calcium antagonism, and inhibition of the inflammatory response [37]. In addition, Carvedilol has been identified as an ideal drug for the treatment of liver fibrosis, with significant survival benefits in patients with cirrhosis [37,38,39,40]. The drug is widely used in clinical practice, and its pharmacokinetics and therapeutic side effects have largely been studied. The aim of our study was to target the delivery of the drug to fully exploit its efficacy. Since Carvedilol can significantly improve MIRI, while reducing inflammation, it can also reduce myocardial cell damage and reduce the impact on blood pressure. It is worth continuing to explore the efficacy of this drug.

Therefore, in our study, we took advantage of the unique biological properties of platelets and wrapped Carvedilol in platelet membranes to reach the site of myocardial ischemic injury by means of blood circulation for treatment (Figure 1). Furthermore, a comparison of intravenous Carvedilol with intraperitoneal administration has demonstrated the safety and feasibility of this targeted delivery method, which is not only more effective but also does not cause significant damage to the organism. This new approach also provides a viable option for the better treatment of clinical patients.

## 2. Materials and Methods

### 2.1. Separation of Platelets and Generation of PMVs

Whole blood from adult SD male rats was collected into EDTA anticoagulant tubes and then centrifuged at 200× *g* for 20 min at room temperature to separate platelet-rich plasma (PRP). PBS containing 5 mM prostaglandin E1 (PGE1, GlpBio, Montclair, CA, USA) was added to the purified PRP to prevent platelet activation. The pellet was then centrifuged at 1800× *g* for 20 min at room temperature and the resulting pellet generated platelets (PLTs). Next, we resuspended the precipitated platelets in PBS containing 5 mM prostaglandin E1. The prepared suspended platelets were counted in a flow cytometer, and the number of platelets per tube was controlled at (2.5 − 3) × 10^5^. To create platelet membrane vesicles (PMVs), we subjected the extracted platelets to repeated freeze–thawing, centrifuged them at 8000× *g* for 5 min, resuspended the cell debris, and then resuspended them for 2 min in a FS30D bath sonicator (Fisher Scientific, Waltham, MA, USA) [41]. After water bath sonication, platelet membranes, i.e., platelet vesicles, were filtered through sterile 0.2 μm filters. PMVs were aliquoted into 1 mL samples and stored unused at −80 °C under the protection of Serum-Free Cell Freezing Medium. Morphological changes of PLTs and PMVs were investigated by SEM (JEM-2100, JEOL Ltd., Tokyo, Japan). In this case, 200 μL of each of the prepared platelet and PMV samples was dropped onto a glass slide. The samples were dehydrated twice with 4% methanol gradient, and the samples were washed with PBS before observation. In addition, the mean particle sizes of both PLTs and PMVs were detected through nanoparticle tracking analysis (NTA).

### 2.2. Preparation and Toxicity Verification of PMVs@Carvedilol

The Carvedilol loading ability of PMVs was measured. Different proportions of PMVs to Carvedilol were used to synthesize the optimized PMVs@Carvedilol drug delivery system. Different masses of Carvedilol were dissolved in equal amounts of saline to prepare different concentrations of the drug (0.125, 0.25, and 0.5 mg/200 μL), and they were encapsulated with PMVs and sonicated at different volume ratios (2:1, 1:1, 1:2, and 4:1). The integrated mixture was centrifuged at 8000× *g* for 5 min, and the drug content of the supernatant was determined using a SpectraMax 190 to record the encapsulation rate and loading rate in order to determine the optimal concentration for drug encapsulation. Since the optimum drug concentration was calculated by assaying the supernatant concentration after centrifugation of the mixture, the effect of precipitated platelet membrane particles on absorbance could be excluded. The encapsulation rate and drug loading rate were calculated as follows:(1)$$\mathrm{encapsulation}\;\mathrm{rate}=\frac{\mathrm{initial}\;\mathrm{concentration}\;\mathrm{of}\;\mathrm{drug}\;-\;\mathrm{actual}\;\mathrm{concentration}\;\mathrm{of}\;\mathrm{supernatant}}{\;\mathrm{initial}\;\mathrm{concentration}\;\mathrm{of}\;\mathrm{drug}}\times 100\%$$
(2)$$\mathrm{drug}\;\mathrm{loading}\;\mathrm{rate}=\frac{\mathrm{mass}\;\mathrm{of}\;\mathrm{drug}\;\mathrm{in}\;\mathrm{encapsulated}\;\mathrm{platelets}}{\mathrm{mass}\;\mathrm{of}\;\mathrm{encapsulated}\;\mathrm{platelets}\;}\times 100\%$$

When the drug concentration was 0.5 mg/200 μL and the volume ratio (drug/PMVs) was 1:1, the best encapsulation efficiency and drug loading rate could be obtained. Therefore, in the following experiments, the concentration of PMVs@Carvedilol was 0.5 mg/200 μL. Carvedilol was diluted with physiological saline. PMVs@Carvedilol were mixed in 1 mL of sterile saline and injected via the tail vein into MIRI rats; in addition, we compared other methods of administration: (1) tail vein injection of 0. 5 mg/mL of Carvedilol; (2) intraperitoneal injection of 0.7 mg/mL of Carvedilol (its concentration was absorbed in vivo with the same efficiency as 0.5 mg/mL). To verify the stability of Carvedilol when encapsulated by PMVs, in vitro experiments were performed. Firstly, PMVs@Carvedilol were prepared and added to a certain amount of normal saline, and the changes in the concentration of the encapsulated drug were observed at different time points (0 h, 2 h, 6 h, 8 h, 12 h, 24 h). The control group received the same concentration of Carvedilol dissolved in the same amount of normal saline, and the dispersion of the drug after membrane encapsulation was observed and compared. To further verify the toxic effects of the platelet-membrane-encapsulated drug in SD rats, we injected PMVs@Carvedilol into the tail vein in normal SD rats and evaluated the changes in several blood components at different time points (1 day, 3 days, 5 days, 7 days). These included inflammatory indicators (neutrophils) and liver and kidney function indicators (ALT, AST, BUN, CREA) to verify that the drug delivery system that we prepared was nontoxic to humans. At least 200 μL of venous blood was drawn from rats and placed in EDTA anticoagulant tubes to measure neutrophils, obtained by an automatic hematology analyzer (MEK-6318K, Nihon Kohden, Tokyo, Japan). At the same time, 50 μL of venous blood was added to each index of liver and kidney function, placed in a heparin tube, and analyzed using an automatic biochemical analyzer (IDEXX Catalyst One^®^, Boxing Biotechnology Co Ltd., Shenzhen, China).

### 2.3. Detection of Platelet Membrane Vesicle-Specific Surface Markers

Cell precipitates or tissue homogenates were lysed with RIPA lysis buffer (NCM Biotech, Suzhou, China, WB3100) and PMSF (Solarbio, Beijing, China, P0100) for 1 h. After centrifugation at 12,000 rpm for 15 min, the supernatant was aspirated, mixed with the loading buffer and incubated at 100 °C for 10 min to obtain protein samples. Western blot experiments using SDS-PAGE Gel electrophoresis were performed. Proteins were detected with the Odyssey Infrared Imaging System (Licor, Lincoln, NE, USA), and the grayscale analysis was performed using ImageJ software. To further confirm successful membrane extract, we performed Western blot analysis by using antibodies against specific platelet surface markers including rabbit anti-rat CD41 (Proteintech, Wuhan, China, 24552-1-AP; 1:1000), activated platelet-specific marker CD62P (Proteintech, 60322-1-Ig; 1:1000), and the intracellular protein GAPDH (Proteintech, 60004-1-Ig; 1:1000).

### 2.4. PMV Target Validation

To verify that PMVs possessed the ability of PLTs to target injury sites in the rat MIRI model, we used Dil dye to label the PMVs for verification. First, we dissolved Dil dye with DMSO, mixed Dil dye at a concentration of 10 μM in PBS-containing PMVs, stained the sample for 20 min, removed excess dye after centrifugation, and resuspended the labeled PMVs in PBS; the entire process was performed in the dark. MIRI models were prepared and labeled PMVs were injected into the tail vein during reperfusion. In the control group, the left anterior descending branch was threaded without ligation. We validated the targeting of platelets on MIRI using the IVIS^®^Lumina XRMS Series III in vivo imager, while we examined the remaining platelets in vivo at different time points 6 h, 8 h, 12 h, and 24 h. This also provided a basis for validating the timing of treatment with PMVs@Carvedilol in vivo.

### 2.5. In Vivo Ischemia/Reperfusion Rat Model

This study was approved by the Ethics Committee of the Affiliated Hospital of Nantong University (No. S20200314-012). Animal care was in line with the Institutional Animal Care and Use Committee (IACUC) guidelines. Preparation of rat myocardial ischemia–reperfusion model: SD rats were anesthetized under isoflurane throughout the surgical procedure. During the operation, the rats were artificially ventilated through a tracheal cannula connected to a small animal ventilator (VentStar Small Animal Ventilator). The rat chest skin was disinfected with iodophor, and the skin was incised approximately 1 cm vertically from the fourth rib on the left side. We used hemostatic forceps to dissociate the subcutaneous tissue and muscle layer-by-layer to fully expose the left 3rd to 4th intercostal space, where the heartbeat can be seen, and then inserted the forceps into the intercostal space. The intercostal space was opened with an intercostal dilator to fully expose the thoracic cavity. The pericardium was carefully cut to determine the location of the left atrial appendage. The left anterior descending coronary artery was ligated with a medical 6–0 thread for 30 min. Perfusion was then resumed, and the thoracic cavity was closed layer by layer [42].

### 2.6. Fluorescence Imaging Analysis

Rat hearts were harvested 24 h after cell infusion and animals were euthanized. Frozen sections were used for immunofluorescence. The slide was first sealed with 5% BSA for 30 min, and then incubated overnight with Bax (Proteintech, 50599-2-Ig; 1:100) and Bcl-2 (Proteintech, 12789-1-AP; 1:100) at 4 °C. Sections were incubated with secondary antibody (1:500) for 1 h at room temperature and nuclei were stained with DAPI (P36935, Life Sciences), followed by encapsulated sections. We placed the hearts in a Xenogen IVIS imaging system (Caliper Life Sciences, Mountain View, CA, USA) and detected the fluorescent signal of the bound antibody. The collected fluorescence images were analyzed and processed by ImageJ.

### 2.7. Cardiac Function Assessment

Cardiac ultrasound was performed in SD rats at 24 h and 1 week after myocardial ischemia–reperfusion to detect cardiac function: rats were anesthetized and placed on an ultrasound thermostat with limbs fixed and connected to an ECG, and cardiac ultrasound was performed with a Vevo 2100 small animal ultrasound machine. The rat’s chest was dehaired, the table was tilted 30° to the left, the precordial region was coated with coupling agent, and the ultrasound probe was placed on the left side of the sternum, pointing at the rat’s left shoulder, to show a short-axis view of the ventricle. Anatomical M-mode ultrasound was used in a short-axis view of the left ventricular papillary muscle, so that the sampling line crossed the anterior septum and the posterior lateral wall. Left ventricular motion curves were recorded and measured based on 2D image-guided M-curves measuring the left ventricular ejection fraction (LVEF%), left ventricular shortening fraction (LVFS%), left ventricular end-diastolic and end-systolic internal diameters (LVDd and LVDs), and left ventricular end-diastolic and end-systolic posterior wall thickness (LVPWd and LVPWs) [43].

### 2.8. TTC Staining

TTC staining procedure: Firstly, 2% TTC solution was prepared; 24 h after the model preparation was completed, the heart was removed, rinsed in saline, sliced into 2 mm sections, and placed in TTC solution for 30 min at 37 degrees to avoid light. The ischemic myocardium was found to be whitish; it was immediately photographed, and the heart slices were fixed with 10% neutral formaldehyde for 6 h. The control group and the other three groups of administration were observed, and the size of the white area was calculated.

### 2.9. Statistical Analysis

The data were evaluated using GraphPad Prism 9.3.0 (GraphPad Software, San Diego, CA, USA) and expressed as mean ± SD. Differences between groups were assessed by one-way analysis of variance and subsequent Tukey post-tests. *p* < 0.05 indicated significance.

### 2.10. Animal Randomization Method

The animal cage exhibited a random order. The method of using paper extraction was physically randomized before the animal experiment. All measurements were carried out in random order, and the surgeons and ultrasound test groups were double-blinded.

## 3. Results

### 3.1. Characterization of Platelet and PMVs In Vitro

The morphology and size distribution of PLTs and PMVs were initially characterized by TEM and nanoparticle tracking analysis (NTA). According to TEM, the extracted PLTs showed characteristic elliptical or circular morphology, while the PMVs showed a complete structure and a more homogeneous size distribution (Figure 2A,B). In addition, we used nanoparticle tracking analysis to analyze the particle size of the extracted PLT and PMVs and found that there was little difference in particle size before and after membrane extraction. We found that the particle size of PLT was mostly concentrated at 167.7 nm, although there was a small peak tail, while the extracted PMV was mostly concentrated at 172.6 nm and was more uniform in size (Figure 2C). Next, to verify the functional integrity of the extracted platelets and their membranes, we examined the platelet membrane-specific marker CD41 (Figure 2D,E), as well as the platelet activation-specific marker CD62P, and the intracellular protein GADPH (Figure 2F,G), where the extracted platelets and platelet membranes well preserved the surface marker CD41, while CD62P was used to verify that the platelets that we extracted were activated. The intracellular protein GD was also used to verify that we were extracting platelet membrane vesicles, removing most of the intracellular material.

### 3.2. Intravenously Injected Platelets Targeting MIRI

During myocardial ischemia–reperfusion injury, endothelial cell injury and inflammatory cell infiltration can lead to platelet (PLT) reaggregation at the site of injury. To verify that the prepared PMVs had PLT targeting the site of ischemic injury to the myocardium, we extracted platelets from the blood of adult SD rats and created PMVs to verify that PMVs could target MIRI. We labeled the extracted PMVs with Dil fluorescent dye and suspended them in PBS solution containing PGE2; then, 30 min after centrifugation, we removed excess dye, and resuspended the Dil-labeled PMVs in PBS. The labeled PMVs were injected into the MIRI group and the control group, respectively, and two sets of fluorescent signals were observed under ex vivo fluorescent imaging for one hour after injection. We found that the MIRI group had a significant infrared fluorescent signal, and the control group had almost no signal (Figure 3A). In order to clarify whether we targeted to the heart area, we removed the two sets of heart tissue, observed them again, and determined whether the red fluorescent signal was able to gather in the MIRI-damaged heart (Figure 3B,C). In order to subsequently ensure deliver the timeliness of the drug delivery, we examined the residual time of the labeled PMVs in the heart at the following time points: 6 h, 8 h, 12 h, and 24 h, respectively (Figure 3D,E). We found the Dil-labeled PMVs remained aggregated on the damaged heart for at least 24 h, which represented the minimum time for the subsequent encapsulation of the drug for treatment in vivo.

### 3.3. Determination of Optimal Drug Encapsulation and Loading Rates

Since Carvedilol has maximum absorption at 285 nm, 319 nm, and 331 nm [44], in order to wavelength, we first prepared standards and diluted Carvedilol with physiological saline to form different concentrations, and we then plotted the curve by enzyme standard detection and found that R^2^ was closest to 1 at 319 nm (Figure 4A), so Carvedilol reached its optimum absorption peak at 319 nm (Appendix A). To determine the optimal encapsulation rate and drug loading rate of PMVs@Carvedilol, we first set three concentrations of Carvedilol (0.125 mg/200 μL, 0.25 mg/200 μL, 0.5 mg/200 μL) with volume ratios (Carvedilol:PMVs) of 2:1, 1:1, 1:2, and 4:1. Ultrasonication was performed. The integrated mixture was centrifuged at 8000× *g* for 5 min and the drug content of the supernatant was determined using the SpectraMax 190 to record the encapsulation and loading rates. When the encapsulation rate was 92% and the drug loading rate was 46% (Figure 4B,C), we could determine that the optimum concentration for encapsulation was 0.5 mg/200 μL at a volume ratio of 1:1, although the actual concentration encapsulated in PMVs was 0.46 mg/200 μL. In order to verify that the concentration of Carvedilol was stably encapsulated within the PMVs, we conducted in vitro experiments to observe the changes in the encapsulated concentration of the drug at different time points (0 h, 2 h, 6 h, 8 h, 12 h, 24 h). The control group received an equal concentration of Carvedilol dissolved in an equal amount of saline. We could observe that with time, the encapsulated Carvedilol concentration decreased slightly from 0.46 mg/200 μL initially to 0.4 mg/200 μL, and the encapsulation rate was essentially stable at more than 80% (Figure 4D). This drug leakage can be disregarded compared to the overall delivery concentration to the injury site. To verify the safety of our prepared platelet membrane vesicles encapsulated with Carvedilol for blood delivery, we injected PMVs@Carvedilol directly into healthy adult SD rats through the tail vein and performed the analysis of blood indicators drawn at 1 day, 3 days, 5 days, and 7 days. The analysis showed no significant differences in inflammatory parameters (neutrophils), liver function parameters (ALT, AST), or kidney function parameters (BUN, CREA) compared to normal uninjected rats (Figure 4E), except for a slight increase in neutrophils on the first day after injection, but this stabilized to normal values in the subsequent time period. This result indicates that our prepared membrane-encapsulated drug has no effect on liver and kidney function, no inflammatory response, and does not affect the MIRI model, demonstrating that it is feasible and safe.

### 3.4. Compared with Other Administration Methods, PMVs@Carvedilol Can Better Reduce Infarct Size, Reduce Myocardial Cell Apoptosis, and Improve Cardiac Function

To test the therapeutic potential of PMVs@Carvedilol, we employed a rat model of MIRI by temporarily ligating the anterior descending coronary artery for 30 min followed by reperfusion, which is the gold standard in the MIRI model. It is the most stable and effective model. In the in vivo experiments, we set up three experimental groups, including (1) intravenous injection of PMVs@Carvedilol, (2) intravenous injection of carvedilol (0.5 mg/mL), and (3) intraperitoneal injection of carvedilol (0.7 mg/mL), a control group (MIRI). To investigate the benefits of the targeted drug therapy, we assessed the cardiac morphology, cytology, and pump function (Figure 5A). Since the therapeutic effect of carvedilol can have both short-term and long-term benefits, we set the observation periods to be 24 h and 1 week. To observe whether Carvedilol reduced the area of myocardial ischemic infarction, we used TTC staining to observe the infarct area in heart sections at 24 h with different administration methods (Figure 5B,C). It could be found that the intraperitoneal injection and intravenously administered medicines had therapeutic effects, but the platelet-delivered drug had a more significant effect. The area of white infarction was relatively reduced, and the highest amount of viable myocardium was observed. In addition to verifying the changes in infarct size under different modalities of treatment, we also measured LVEF%, LVFS%, LVDd, and LVDs, indicators of cardiac function in rats, by performing small animal ultrasound (Figure 6A–E). The indicators of cardiac ultrasound can show very visually the short and long-term therapeutic effects on cardiac function in rats under treatment. We could observe that intraperitoneal and direct intravenous administration did have a therapeutic effect, but the therapeutic effect of the intravenous platelet membrane-loaded drugs was significantly higher than in the other two treatment groups, indicating that the targeted administration of the drug is feasible and significantly more effective. Apoptosis is regulated by a variety of genes. Among a series of apoptosis-regulating genes involved in apoptosis, the Bcl-2 gene and the BAX gene of the same family are important apoptosis-regulating genes and are considered to be one of the last common pathways in the regulation of apoptosis. Bcl2, also known as an antiapoptotic protein, inhibits cell death caused by a variety of cytotoxic factors and reduces the production of oxygen free radicals and the formation of lipid peroxides. The extent of myocardial apoptosis that can result from MIRI can be detected with Bcl2 antibodies as an indicator of myocardial injury (Figure 7A,B). BAX, as a proapoptotic antibody, can be used as one of the indicators of apoptosis detection in cardiac myocytes. We used Bcl2 and BAX antibodies to stain heart sections under different groups and observed the apoptosis of cardiomyocytes by immunofluorescence (Figure 7C,D). Staining cardiomyocytes with Bcl2 and BAX, and nuclei with DAPI, we could observe that the red fluorescence of Bcl2 was significantly stronger in the group receiving platelet membrane-targeted therapy compared to the other groups, and the green fluorescence of BAX proapoptosis was significantly weaker than in the other groups. This suggests a better effect of the platelet membrane delivery of Carvedilol in reducing cardiomyocytes apoptosis compared to the other three groups. Various metrics validating cardiac therapy have shown that the absorption rate of intravenous drug delivery is already slightly higher than that of intraperitoneal or oral drug delivery. Thus, when we deliver the drug in a targeted manner, we can treat the condition with a lower concentration of drug delivery but targeting to the site of injury can increase the absorption rate of the drug, improve therapeutic efficacy, and reduce the loss of the drug by other means of drug delivery.

## 4. Discussion

In the present study, we applied the natural injury-targeting power of platelets to enhance pump function and reduce the infarction area by applying Carvedilol. Our data showed that: (1) PMVs retain platelet membrane proteins, remove intracellular proteins, and retain the ability of myocardial injury targeting. (2) We determined the best encapsulation rate and drug rate of PMVs-encapsulated Carvedilol. In addition, we performed a safety assessment of the membrane-encapsulated drug system in vivo. We confirmed that our prepared membrane-encapsulated Carvedilol had no effect on liver or kidney function and no inflammatory response. (3) PMV modification enhances the reduction in cardiac function caused by Carvedilol to treat ischemia–reperfusion injury.

Myocardial ischemia–reperfusion injury is one of the most studied conditions among ischemic coronary artery disease, and the mechanisms of injury have been widely explored. As the damage to myocardial cells is irreversible, much of the treatment aims to control further damage to myocardial cells as well as cardiac function. There are a number of treatment options currently applied in the clinic, including the initial drug therapy, surgical interventions, and the now widely researched cellular therapy. However, the greatly reduced efficacy of drugs, the further trauma of surgical interventions and even the unknown long-term risks of cellular therapies are all obstacles to the improved treatment of MIRI and are areas that need to be investigated. In recent years, new cell therapies have focused on better reducing myocardial cell damage and restoring heart function. For example, pluripotent stem cell-derived cardiomyocytes are modified into gel dressings or injected directly into the heart for treatment. In addition to exploring stem cells and secreted exosomes, researchers have also uncovered additional potential therapeutic factors. In order to determine how these cells and factors, which have better therapeutic potential, can be brought to the site of injury with less trauma, researchers have identified a class of circulating cells that have the ability to circulate to the targeted site of injury, with platelets being a prevalent topic of recent research. This is because they have many functions, such as aggregating at the site of vascular damage, participating in the immune response and being involved in the tumor process.

Thus, understanding the mechanism by which platelets target MIRI will help us in better understanding the present experiment. Firstly, platelets have the capacity for passive targeting, mainly because there are many receptors on the surface of the platelet membrane, including CD47, which has platelet autoantigens and can be protected from recognition and attack by the organism with camouflage. When vascular endothelial cells are injured, in addition to vasoconstriction, it also activates platelet adhesion and aggregation to the injury site; secondly, platelets have the capacity for active targeting, mainly because their membrane surface proteins, such as P-selectin, have many specific functions, and they can bind specifically to ligands on the surfaces of tumor cells, or to ligands specific to injured vascular endothelial cells, etc. Thus, this novel therapeutic modality using platelet membranes for drug delivery has great therapeutic potential in vascular diseases and tumor microenvironments.

Currently, there are two types of calcium channel blockers, selective and nonselective, among which Carvedilol has α1 receptor blocking and nonselective β receptor blocking effects, which cause capillary dilation, thus reducing peripheral resistance and lowering blood pressure. Currently, Carvedilol is mainly used to treat heart failure and lower hypertension, so we prepared a novel drug delivery system to verify its therapeutic effect in MIRI [45,46].

In our study, unlike other studies, we used PMVs directly for encapsulation without excessive modification, thus reducing the potential toxicity of the modified structure to the organism; we were the first to start using PMVs directly for the delivery of Carvedilol in the study of MIRI. We overcame this and also evaluated the stability of the encapsulated drug concentration. We tested the encapsulated drug in vitro for a period of time and found no significant leakage of the encapsulated drug, which confirmed that Carvedilol was essentially encapsulated in the PMVs. Thus, our study combined drug therapy with novel cellular therapies to fully exploit the efficacy of the drug in reducing myocardial injury while avoiding drug wastage.

In this paper, we have much to refine; for example, as PMVs@Carvedilol circulates in the blood, in addition to verifying the long-term safety in vivo, it is necessary to consider the metabolism of the drug in vivo and the effects of the drug on other organs, which is an aspect that we need to refine in subsequent experiments. It is important not to overlook the possible effects of the slight elevation of neutrophils on the first day of injection when examining the toxicity of PMVs to the organism. Furthermore, it is unknown whether this type of drug encapsulation within the membrane is applicable to other drug treatments—there are mechanisms regarding drug encapsulation within the membrane that need to be further explored.

This introduces new ideas for our subsequent experiments and provides a more comprehensive insight into platelets and even other circulating cell membranes or cell membranes.

## 5. Conclusions

In conclusion, we demonstrate a novel therapeutic approach for the preparation of platelet membrane vesicle-encapsulated drugs that retain the natural targeting ability of platelets and the biological function of the Carvedilol drug. As a targeted therapy, PMVs@Carvedilol preferentially accumulate in endothelial cells during ischemic heart injury, while enhancing the ability of Carvedilol to attenuate cardiomyocyte apoptosis, improve infarct size, and improve cardiac function in a MIRI rat model. Therefore, as a novel approach, PMVs @Carvedilol may provide a safer and more effective treatment as a novel approach, which can also be used in the treatments with other biomimetic drugs.

## Figures and Tables

**Figure 1 membranes-12-00605-f001:**
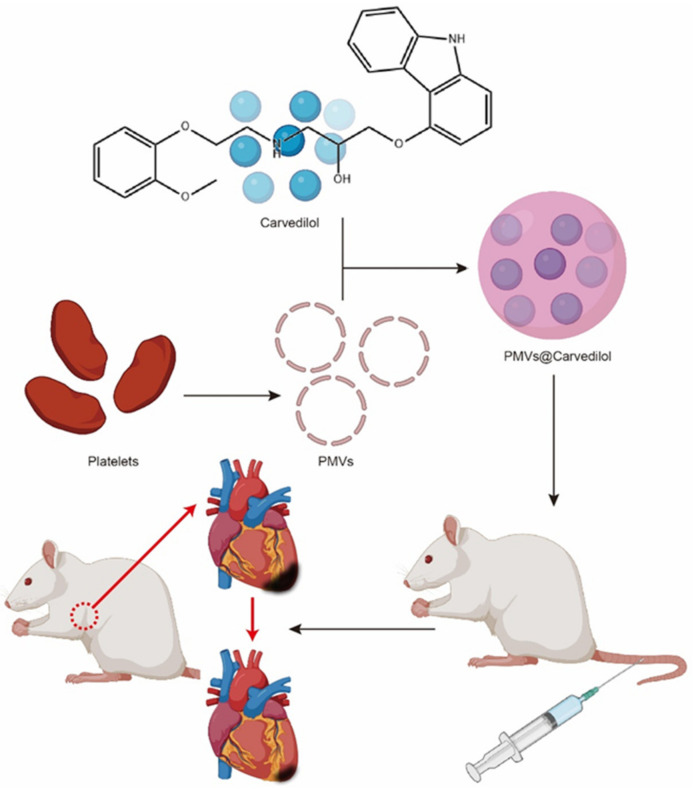
Schematic diagram of PMVs@Carvedilol fabrication and its targeted therapy towards myocardial ischemia–reperfusion injury.

**Figure 2 membranes-12-00605-f002:**
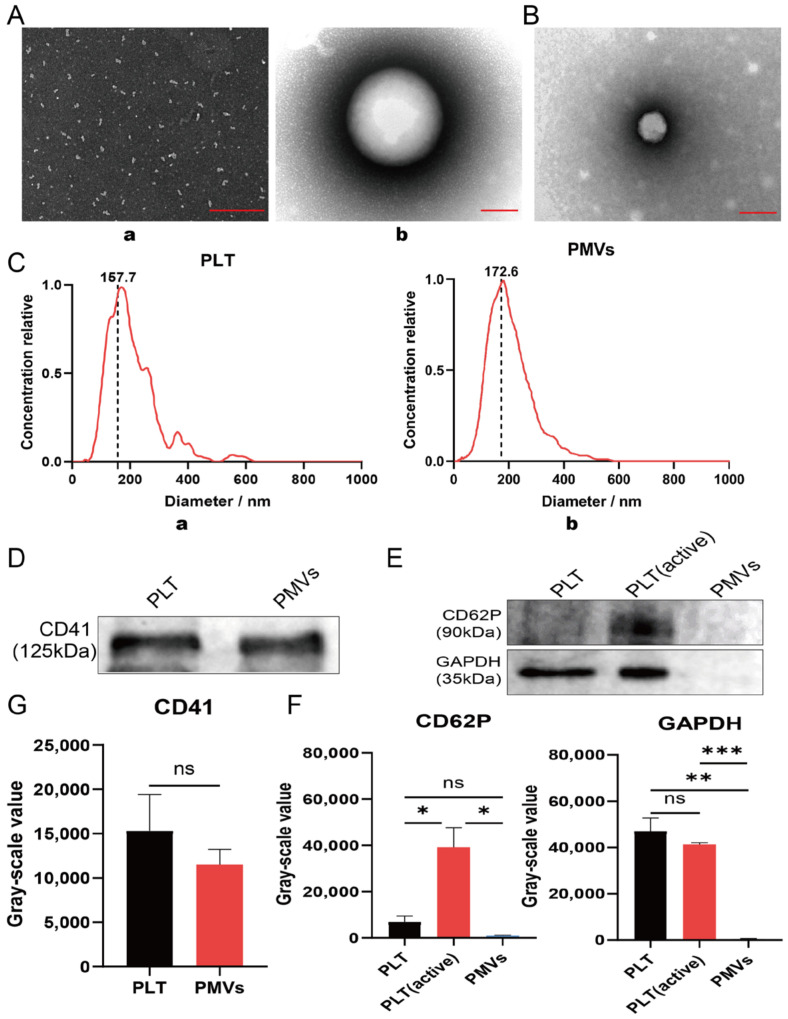
Characterization of PLT and PMVs in vitro. (**A**) SEM image of PLT. **a**—Scale bar, 1.0 μm; **b**—Scale bar, 100 nm. (**B**) A transmission electron micrograph of platelet membrane vesicles (PMVs). Scale bar, 100 nm. (**C**) **a**,**b**—Size distribution of PLT and PMVs by nanoparticle tracking analysis. (**D**) Western blot analysis revealed the expression of platelet-specific markers including CD41 in platelets, PMVs. (**E**) Quantitative analysis of the membrane protein CD41. (**F**) CD62P and GAPDH expression levels of PLT, PLT (active), and PMVs (PLTs were activated with 1% CaCl_2_ for 2 h). (**G**) Quantitative analysis of CD62P and GAPDH. * *p* < 0.05; ** *p* < 0.01; *** *p* < 0.001; ns > 0.05.

**Figure 3 membranes-12-00605-f003:**
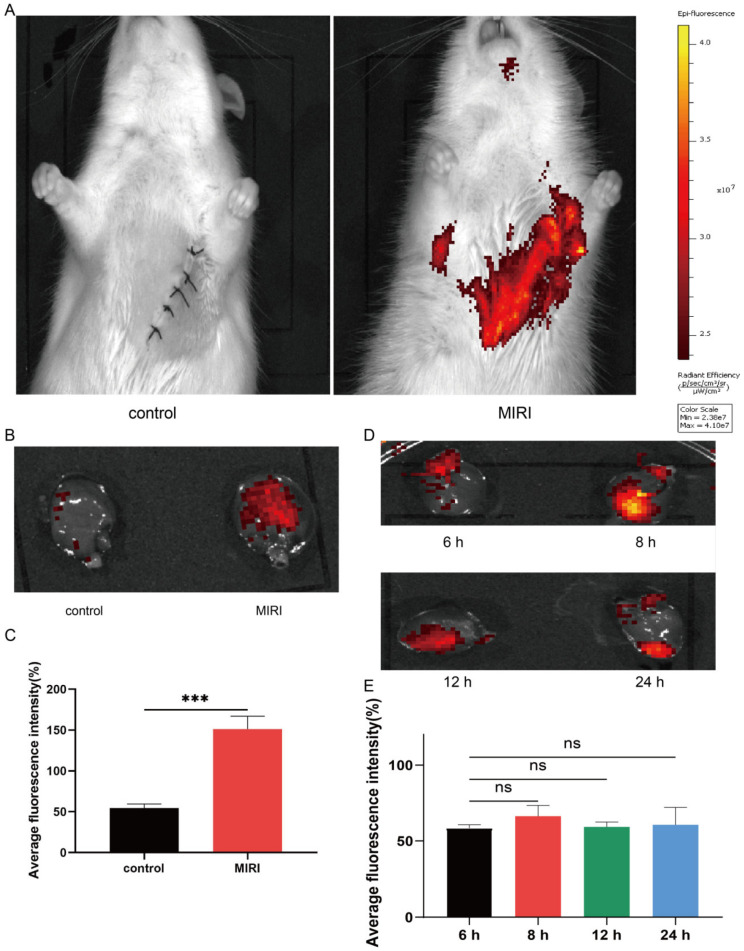
Verification of PMV targeting in vivo. (**A**) A representative in vitro fluorescence image showing intravenous Dil-labeled PMV binding in hearts with or without MIRI. (**B**,**C**) Representative fluorescent image showing the targeting of Dil-labeled PMVs (red) to the MIRI area and quantitative fluorescence analysis. (**D**,**E**) Retention time and fluorescence quantification of fluorescent PMVs in the MIRI rat heart. * *p* < 0.05; ** *p* < 0.01; *** *p* < 0.001; ns > 0.05.

**Figure 4 membranes-12-00605-f004:**
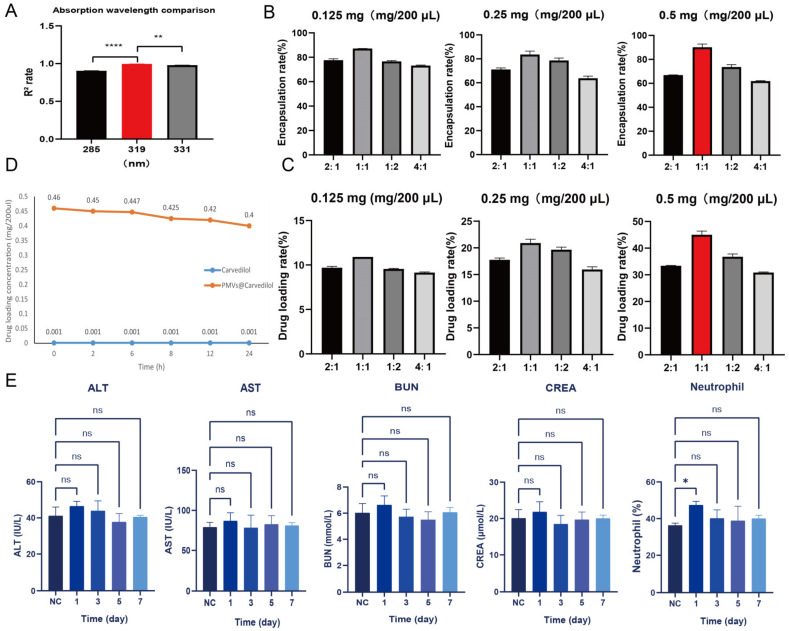
Determination of optimal drug encapsulation and loading rate for PMVs@Carvedilol and validation of toxicity in vivo. (**A**) Comparison of the maximum absorption of Carvedilol at 285 nm, 319 nm, 331 nm. (**B**) Comparison of the encapsulation rate at different concentrations and volume ratios. (**C**) Comparison of the drug loading rate at different concentrations and volume ratios. (**D**) Changes in the encapsulation concentration of PMVs@Carvedilol at different time points (0 h, 2 h, 6 h, 8 h, 12 h, 24 h). (n = 3) (**E**) Analysis of various blood parameters of normal rats at different time periods after tail vein injection of PMVs@Carvedilol. * *p* < 0.05; ** *p* < 0.01; **** *p* < 0.0001.

**Figure 5 membranes-12-00605-f005:**
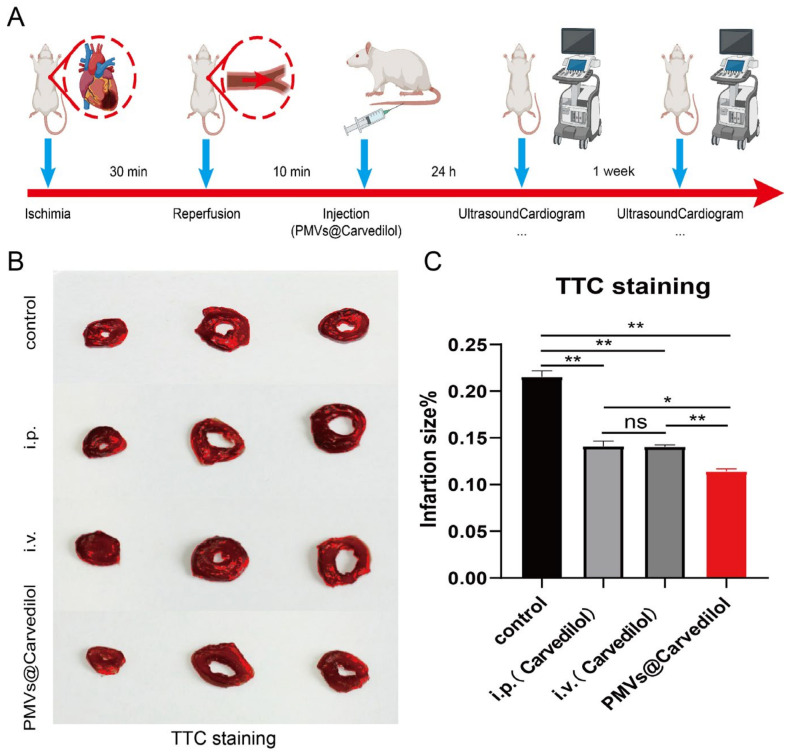
PMVs encapsulation reduces infarction area and boosts therapeutic outcomes in rats with MIRI. (**A**) A schematic showing the animal study design. (**B**) Representative TTC-stained myocardial sections 24 h after treatment (red fluorescence, scar tissue; red, viable myocardium). Scale bar, 2 mm. (**C**) Quantitative analyses of viable myocardium and scar size from the TTC staining images. (i.p. means intraperitoneal injection; i.v. means intravenous injection.) * *p* < 0.05; ** *p* < 0.01; ns > 0.05.

**Figure 6 membranes-12-00605-f006:**
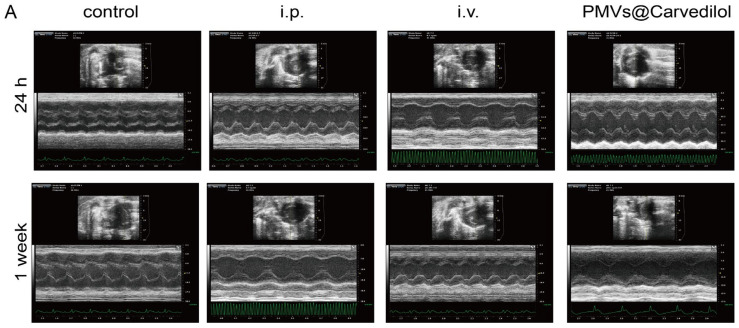
Detection of ultrasound cardiogram in rats from different groups. (**A**) Echocardiography in SD rats 24 h and 1 week after MIRI. (**B**) Quantitative analyses of 24 h heart LVEF% and LVFS%. (**C**) Quantitative analyses of 1 week heart LVEF% and LVFS%. (**D**) Quantitative analyses of 24 h heart LVDd and LVDs. (**E**) Quantitative analyses of 1w heart LVDd and LVDs. (i.p. means intraperitoneal injection; i.v. means intravenous injection.) * *p* < 0.05; ** *p* < 0.01; *** *p* < 0.001; **** *p* < 0.0001.

**Figure 7 membranes-12-00605-f007:**
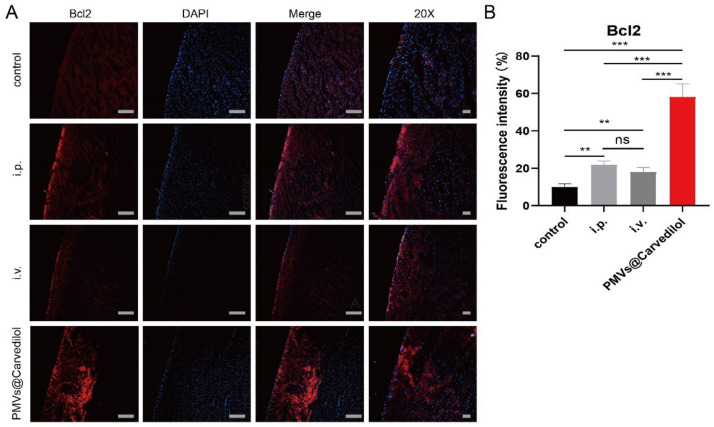
PMVs@Carvedilol therapy reduces myocyte cell apoptosis. (**A**) Expression of antiapoptotic molecule Bcl2 under immunofluorescence. Scale bar, 600 μm; 20× Scale bar, 150 μm. (**B**) Quantitative analyses of antiapoptotic molecule Bcl2. (**C**) The expression of apoptotic molecules Bax under immunofluorescence. Scale bar, 600 μm; 20× Scale bar,150 μm. (**D**) Quantitative analyses of apoptotic molecule Bax. (i.p. means intraperitoneal injection; i.v. means intravenous injection.) ** *p* < 0.01; *** *p* < 0.001; **** *p* < 0.0001.

## Data Availability

The data presented in this study are available on request from the corresponding author.

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
