# Peer review of "Platelet-Membrane-Encapsulated Carvedilol with Improved Targeting Ability for Relieving Myocardial Ischemia–Reperfusion Injury"

_membranes, 2022, doi:10.3390/membranes12060605_

Round 1
Reviewer 1 Report
The manuscript by Zhou et al. describes the loading, characterization, and in vivo testing of platelet membrane derived nanostructures loaded with carvedilol. The manuscript is reasonably well written, and results seems interesting. Since my main expertise is related to assembly and characterization of the nanostructures, I will focus the review on this part, leaving the final decision on in vivo evaluation (that seems complete to me) to other reviewer's judgement.
Concerning the formulation of the drug vehicle, I see a couple of issues that should be addressed before publication:
1) While I understand the process of formulation of the nanostructures, in vitro leakage test is not presented. It would be important to provide evidences that the drug is actually internalized in the nanostructures and there is no leakage. Moreover, it is not clear whether the drug is actually adsorbed, at least to some extent, on the membrane. The authors should evaluate leakage profile during time in simulated conditions (e.g. dispersion in saline)
2) the authors use the term "concentration" referring to what seems to be an amount (since it is expressed in mg) or at most a mass ratio. Please check
3) related to issue 2, the actual concentration (mg/mL) at which the loading procedure is performed should be clearly indicated, since it can have an impact on loading efficiency. Did the authors test loading at different concentrations to determine whether there is an impact on loading efficiency?
Reviewer 2 Report
The manuscript by Zhou et al. addressed the use of platelet-derived membrane vesicles as carriers for the delivery of carvedilol in a model of myocardial ischemia-reperfusion in rats. The authors found a positive effect in terms of cardiac function in the treated group. The conclusions are mostly supported by the data. However, the manuscript needs major changes in terms of presentation, in addition, some aspects require further clarification:
1- The whole manuscript needs to be checked in terms of grammar and syntax. Several parts of the manuscript need to be reformulated in order to improve the readability.
2- The concentration of carvedilol is not clear. Currently, this is presented as, for example, “0.125; 0.25 and 0.5 mg/200 g”. It is not clear which amount corresponds to carvedilol or in which volume these amounts were dissolved. Please, clarify
3- Nanoparticle tracking analysis and electron microscopy need to be described in materials and methods.
4- Section 2.4. It is not clear whether platelets were labelled and injected or platelet-derived vesicles. This needs to be specified in M&M as well as in the results section. Furthermore, the process of labelling with Dil dye needs to be further described.
5- Currently, caspase is analysed by fluorescence microscopy. Further quantitative studies are required (e.g., western blot for cleaved PARP). In addition, the analysis of the microscopy images needs to be described in M&M.
6- Section 2.6. The word “nucleofections” should be replaced by “nuclei”.
7-The methods for the quantification of ALT, AST, CREA, BUN and neutrophils need to be described in M&M.
8-The statement on the tolerability or lack of toxicity of the platelet-derived particles needs to be softened throughout the manuscript. Only studies at short-term were performed. In addition, the changes in the neutrophils at day 1 cannot be neglected.
9-Figure 5C. Indicate the drug injected in the second and third experimental groups.
Reviewer 3 Report
The manuscript describes the use of platelet membrane-encapsulated Carvedilol for relieving myocardial ischemia-reperfusion injury. The aim is clear and the strategy interesting. However, several features in the text are unclear, and the English need a string and accurate revision.
Major points
Figure 2B quality is insufficient. The morphology of vesicles appears unusual and typical of an artifact. Better images should be provided.
In the Introduction the Introduction to the background of the use of platelets as nanovectors is insufficient. Were they used in the past? When? With what results? The information provided is not exhaustive and there is need to provide a clear, although brief, picture of the state of the art
In the Introduction, the biological model organism used for the delivery of Carvedilol should be introduced immediately to readers
He Discussion is too long an contains too many details recapitulating the Results section and not discussing the Results implications.
Minor points
There are several sentences that appears not correct and the English need to be fixed, as for example in the Abstract “the efficacy of PMVs encapsulated drug targeted delivery treatment was observed.”, in the Introduction “Platelet adheres to coronary endothelial cells cause acute thrombosis…”, in Results 3.1 “…in a circular shape with structural integrity and the extracted membrane morphology was not disrupted”
Other sentences are unclear, as for example in the Introduction “Further, the treatment of various diseases can be targeted by platelet membrane delivery drug or binding molecules or the like, and the treatment efficacy of the drug is improved [15- 17].” and “Therefore, for the action of platelet targeted ischemic myocardium, we extract platelets, use their thin film function and drug delivery to target damage to achieve the treatment effect [16].” should be reformulated. In Results 3.2 the meaning of “we use the concentration gradient to absorb spectrometry under the enzyme gauge” and “detect the supernatant by an enzyme analyzer” is difficult to catch.
Avoid repetition, as in Results 3.2 “It is determined that the best package concentration and volume are determined.”
In Results 3.2, authors should anticipate (with respect to Results 3.4) and explain the main features of the MIRI model and why this was chosen.
In Results 3.3, it is explained that the “experimental group is divided into three groups”, but the sentence is confused and to understand the three groups the reader has to search elsewhere.
Round 2
Reviewer 1 Report
The authors addressed my concerns. The manuscript is now, in my opinion, suitable for publication
Reviewer 2 Report
The authors have addressed all my concerns.
Reviewer 3 Report
Authors addressed the issue raised in the first version of the manuscript, which has improved its weaknesses